# Association of Metabolic Syndrome with Sensorineural Hearing Loss

**DOI:** 10.3390/jcm10214866

**Published:** 2021-10-22

**Authors:** Hwa-Sung Rim, Myung-Gu Kim, Dong-Choon Park, Sung-Soo Kim, Dae-Woong Kang, Sang-Hoon Kim, Seung-Geun Yeo

**Affiliations:** 1Department of Otolaryngology—Head & Neck Surgery, School of Medicine, Kyung Hee University, Seoul 02454, Korea; marslover@naver.com (H.-S.R.); kkang814@naver.com (D.-W.K.); hoon0700@naver.com (S.-H.K.); 2Department of Otorhinolaryngology, Samsung Changwon Hospital, Sungkyunkwan University School of Medicine, Changwon 51353, Korea; mgent.kim@samsung.com; 3St. Vincent’s Hospital, The Catholic University of Korea, Suwon 16247, Korea; dcpark@catholic.ac.kr; 4Department of Biochemistry and Molecular Biology, School of Medicine, Kyung Hee University, Seoul 02447, Korea; sgskim@khu.ac.kr

**Keywords:** metabolic syndrome, sensorineural hearing loss

## Abstract

The prevalence of sensorineural hearing loss has increased along with increases in life expectancy and exposure to noisy environments. Metabolic syndrome (MetS) is a cluster of co-occurring conditions that increase the risk of heart disease, stroke and type 2 diabetes, along with other conditions that affect the blood vessels. Components of MetS include insulin resistance, body weight, lipid concentration, blood pressure, and blood glucose concentration, as well as other features of insulin resistance such as microalbuminuria. MetS has become a major public health problem affecting 20–30% of the global population. This study utilized health examination to investigate whether metabolic syndrome was related to hearing loss. Methods: A total of 94,223 people who underwent health check-ups, including hearing tests, from January 2010 to December 2020 were evaluated. Subjects were divided into two groups, with and without metabolic syndrome. In addition, Scopus, Embase, PubMed, and Cochrane libraries were systematically searched, using keywords such as “hearing loss” and “metabolic syndrome”, for studies that evaluated the relationship between the two. Results: Of the 94,223 subjects, 11,414 (12.1%) had metabolic syndrome and 82,809 did not. The mean ages of subjects in the two groups were 46.1 and 43.9 years, respectively. A comparison of hearing thresholds by age in subjects with and without metabolic syndrome showed that the average pure tone hearing thresholds were significantly higher in subjects with metabolic syndrome than in subjects without it in all age groups. (*p* < 0.001) Rates of hearing loss in subjects with 0, 1, 2, 3, 4, and 5 of the components of metabolic syndrome were 7.9%, 12.1%, 13.8%, 13.8%, 15.5% and 16.3%, respectively, indicating a significant association between the number of components of metabolic syndrome and the rate of hearing loss (*p* < 0.0001). The odds ratio of hearing loss was significantly higher in subjects with four components of metabolic syndrome: waist circumference, blood pressure, and triglyceride and fasting blood sugar concentrations (*p* < 0.0001). Conclusions: The number of components of the metabolic syndrome is positively correlated with the rate of sensorineural hearing loss.

## 1. Introduction

Sensorineural hearing loss is an important public health problem whose prevalence has increased as life expectancy has become longer [1]. For example, the Global Burden of Disease Study reported that the prevalence of hearing loss increased from 14.33% in 1990 to 18.06% in 2015 and that hearing loss is the fifth most frequent cause of disability in both developed and developing countries [2]. Approximately 50% of people aged over 70 years and 80% of those aged over 85 years experience hearing loss, which affects their ability to communicate and their social lives [1]. The pathophysiological mechanisms of hearing loss are complex, although several risk factors have been reported to contribute to hearing loss, including genetic factors, inflammatory processes, systemic diseases, noise, medications, oxidative stress, and aging [3].

Metabolic syndrome is a disease that includes hypertension, central obesity, hyperlipidemia, and diabetes [4]. Metabolic syndrome was found to be associated with various clinical conditions, including stroke, myocardial infarction, death from cardiovascular disease, and diabetes [5,6]. Studies have recently reported that metabolic syndrome may be associated with hearing loss [3,7,8,9,10,11,12]. Using health examination data, this study therefore investigated the relationships between metabolic syndrome and hearing loss. In addition, studies published on this relationship were subjected to comparative analysis and a systematic review.

## 2. Materials and Methods

### 2.1. Study Population

This study included a total of 94,223 people, ranging in age from their 20s to their 70s, who underwent health checkups, including hearing tests, at a tertiary university hospital from January 2010 to December 2020. Height, weight, body mass index, waist circumference and blood pressure (both systolic and diastolic) were measured in all subjects, and all subjects underwent hearing and blood tests and pulmonary function tests (PFT). Subjects without hearing test results and those with a ≥ 20 dB difference between the two hearing thresholds, a history of surgery for otitis media, or suspected central disease were excluded. The study protocol was approved by the Institutional Review Board of Kyung Hee University Medical Center (KMC 2019-07-065).

### 2.2. Definition of Metabolic Syndrome

The criteria for metabolic syndrome were defined according to the revised National Cholesterol Education Program Adult Treatment Panel III: [13] (1) waist circumference (WC) > 90 cm for men or > 80 cm for women, (2) fasting blood sugar > 100 mg/dL or a diagnosis of diabetes, (3) BP >130/85 mmHg or a diagnosis of high blood pressure, (4) triglyceride concentration > 150 mg/dL, and (5) high-density lipoprotein-cholesterol (HDL-C) < 40 mg/dL for men or < 50 mg/dL for women. Subjects with three or more of these five factors were defined as positive for metabolic syndrome.

### 2.3. Hearing Test

Hearing threshold tests were performed at frequencies of 500, 1000, 2000, 3000, 4000, and 6000 Hz, measured in that order, followed by a retest at 1000 Hz, with air conduction measured at a frequency of 500 Hz. Pure tone audiometry was performed using the standard hexadecimal method, by measuring the air conduction values of each ear at four frequencies (500, 1000, 2000, and 4000 Hz) and calculating their sum using the equation: (500 Hz + 2 × 1000 Hz + 2 × 2000 Hz + 4000 Hz)/6 [14,15,16,17]. 

### 2.4. Research on Hearing Loss and Metabolic Syndrome

To systematically review the results of studies on the correlation between hearing loss and metabolic syndrome, the Scopus, Embase, PubMed, and Cochrane library databases were searched using the keywords “hearing loss” and “metabolic syndrome”. A total of 1373 papers were identified initially, including 390, 612, 364, and seven, respectively, in these databases. Of these, 1300 papers were excluded due to lack of relevance in the title, and an additional 44 papers were excluded due to duplication of papers. Of the 29 papers reviewed in full, 17 were included in the systematic analysis (Table 1).

### 2.5. Statistics

Subjects were divided into two groups, those with and without metabolic syndrome, for analysis of clinical results, and into six groups by age, 20s, 30s, 40s, 50s, 60s, and 70s and over, for analysis of average hearing threshold and degree of hearing loss. To evaluate risk factors related to hearing loss, subgroups were classified according to the number of factors corresponding to the diagnostic criteria for metabolic syndrome. Continuous variables were compared by t-tests and categorical variables by chi-square tests and Fisher’s exact tests. Normal and adjusted odds ratios (ORs) of the relationships of hearing loss with age and number of factors of metabolic syndrome were analyzed by univariate and multivariate logistic regression analyses. All statistical analyses were performed using IBM SPSS version 22 (IBM Corp., Armonk, NY, USA), with *p* values < 0.05 defined as statistically significant.

## 3. Results

Of the 94,223 subjects, 11,414 (12.1%) had metabolic syndrome and 82,809 (87.9%) did not. The mean ages of these two subgroups were 46.1 and 43.9 years, respectively. Of the 11,414 subjects with metabolic syndrome, 8255 (72.3%) were men, whereas, of the 82,809 subjects without metabolic syndrome, 45,200 (54.5%) were men, making the percentage of men significantly higher in subjects with metabolic syndrome (*p* < 0.001). Waist circumference, BP, and triglyceride and fasting blood glucose concentrations were all significantly higher in subjects with metabolic syndrome than in subjects without it. (*p* < 0.001) (Table 2).

A comparison of hearing thresholds by age in subjects with and without metabolic syndrome showed that the average pure tone hearing thresholds were significantly higher in subjects with metabolic syndrome than in subjects without it in all age groups. (*p* < 0.001) (Figure 1). In addition, comparisons by age group confirmed that the percentages of subjects with hearing loss were higher in subjects with metabolic syndrome than in subjects without metabolic syndrome, and there was a significant statistical difference in the groups in their 30s, 40s, and 50s. (*p* < 0.05) (Table 3).

Subjects in this study were divided into subgroups according to the number of components belonging to the diagnostic criteria for metabolic syndrome, and the proportions of subjects diagnosed with hearing loss in each group were compared. Rates of hearing loss in subjects with 0, 1, 2, 3, 4, and 5 of the components of metabolic syndrome were 7.9%, 12.1%, 13.8%, 13.8%, 15.5%, and 16.3%, respectively, indicating a significant association between the number of components of metabolic syndrome and the rate of hearing loss. (*p* < 0.0001) (Table 4).

To correct for the effects of age and gender on the relationship between BMI (Body Mass Indx) and hearing loss, multivariate analysis was performed to determine the normal and corrected odds ratios of hearing loss. The odds ratio of hearing loss was 1.89-fold higher in women than in men, and it was 1.13-fold higher as age increased by 1 year (*p* < 0.0001). Relative to subjects with 0 components of metabolic syndrome, the odds ratios of hearing loss in subjects with 3, 4, and 5 components of metabolic syndrome were 1.06 (*p* = 0.1119), 1.29. (*p* < 0.0001), and 1.21 (*p* = 0.1410), respectively (Table 5).

When analyzing the influence of each component of metabolic syndrome, the odds ratio of hearing loss was significantly higher in the group with four factors: waist circumference, blood pressure and triglyceride, and fasting blood sugar concentrations (*p* < 0.0001) (Table 6).

## 4. Discussion

This cross-sectional study involving a large population assessed the relationship between metabolic syndrome and hearing loss. This study is based on health examination data for relatively healthy patients. In this study, 12.1% or 11,413 subjects met the diagnostic criteria for metabolic disease. First, due to the nature of this study, we cannot rule out the possibility that the study population was biased toward people interested in health. In addition, it is possible that there were fewer subjects who met the criteria for metabolic syndrome because at the time of assessment subjects may have been taking medications to control blood pressure, blood sugar, and/or other blood levels. However, the results of this study are meaningful because sufficient number of subjects, 94,223 was included.

To compare the incidence of hearing loss in adults with and without metabolic syndrome after adjustment for age, the rates of hearing loss were analyzed in subjects with 3, 4, and 5 of the components of metabolic syndrome, and the factors most closely related to hearing loss were identified.

The result of this study demonstrated an association between hearing loss and metabolic syndrome. The threshold of PTA was significantly higher in the group with metabolic syndrome in Table 2, and the rate of hearing loss was also higher in the group with metabolic syndrome in Table 3. In addition, Table 5 shows that although statistical significance was not observed for 3 and 5 factors in the multivariable analysis, this can be attributed to a decrease in the number of relevant groups, and the odds ratio of hearing loss tends to increase according to the number of metabolic syndrome factors. In particular, after adjustment for other factors, four specific components of metabolic syndrome—waist circumference, blood pressure, and triglyceride and fasting blood glucose concentrations—were strongly associated with hearing loss.

Although the mechanism(s) underlying the correlation between metabolic syndrome and hearing impairment remain undetermined, peripheral vascular disorders may be associated with both of these conditions.

Analysis of data from the US National Nutrition Survey showed a link between hearing loss and high blood pressure [28]. Hypertension causes hemorrhage in the inner ear, leading to reductions in capillary blood flow and oxygen supply and resulting in progressive or sudden sensorineural hearing loss [29]. Hypertension also reduces blood flow by reducing the inner diameter of blood vessels in the inner ear through atherosclerosis [30].

Diabetes is associated with microvascular and neuropathy complications affecting the retina, kidneys, peripheral arteries, and peripheral nerves [31,32]. Pathological changes in diabetes can cause sensorineural hearing loss through damage to the blood vessels and nervous system of the inner ear. In autopsy studies of diabetic patients, changes in labyrinth artery, spiral ganglion, cochlear blood vessels, and cranial nerve 8 were observed [33]. Correlations have been observed between diabetes and hearing loss, [19,21,24] and high fasting blood glucose concentrations were found to be independently associated with hearing loss [7,34].

Recent studies have shown that increases in triglyceride concentrations are closely associated with reduced hearing function [35]. Vacuolar edema and degeneration of the vascular striatum were reported as pathological changes associated with dyslipidemia in guinea pigs fed a lipid-rich diet [36,37]. Decreased nitric oxide production and increased reactive oxygen levels due to dyslipidemia can lead to hearing damage [22,37,38,39]. Furthermore, HDL was shown to have anti-inflammatory, antioxidant, and anti-apoptotic effects that can attenuate the pathological changes caused by dyslipidemia [34,40] High TG/HDL ratio has been associated with hearing loss [23] as has low HDL [3].

Obesity has been associated with hearing loss in both humans and animals. Central obesity, increased waist circumference, and increased hearing threshold after BMI correction of the content of visceral adipose tissue were found to be related [20]. Obesity, as determined by BMI, has been associated with an increased risk of hearing loss [26]. Moreover, abdominal lipid-related factors were reported to be associated with hearing loss at specific frequency bands [25]. Abdominal adipose tissue has also been associated with hearing loss in women, [8] and weight-hip ratio (WHR) may be an indicator of the risk of hearing loss [18].

Seventeen previous studies have examined the relationship between metabolic diseases and hearing loss. Nine of these studies analyzed the correlation between the overall components of metabolic syndrome and hearing loss; four analyzed the correlations between indices such as BMI, WHR, and factors relevant to abdominal fats (FRA) and hearing loss; three analyzed the correlations of diabetes DM and HbA1c concentration with hearing loss, and one analyzed the correlation of TG/HDL ratio with hearing loss. Of these 17 studies, 12 were cross-sectional in design, two were prospective studies, two were retrospective studies, and one as a case-control study. Most of the study subjects were middle-aged.

One study reported that metabolic syndrome was related to high-frequency hearing loss in a noise-exposed population [27]. Two studies found that metabolic syndrome was related to hearing loss in elderly women [8,9]. Three studies reported that the percentage of subjects with hearing loss increased as the number of diagnostic factors for metabolic syndrome increased, with the rate of hearing loss being high in patients with four or five factors [3,10,11]. A study in drivers from West Azerbaijan showed the importance of analyzing each component of metabolic syndrome, not metabolic syndrome itself, finding a correlation between each component of metabolic syndrome and hearing loss [12]. One study found that metabolic syndrome itself was not an independent risk factor for hearing loss; rather, increased fasting plasma glucose concentration was the only independent risk factor for hearing loss [22]. In comparison, another study found that metabolic syndrome itself and several of its specific components, such as central obesity, hyperglycemia, and low HDL, were positively associated with hearing loss [7].

Many studies have found a correlation between HDL and hearing loss. For example, one study in US adults found that low HDL correlated with hearing loss, with low HDL being most responsible for the correlation between hearing loss and metabolic syndrome. [3] Two studies in the Korean population reported that low HDL and high TG were highly correlated with hearing loss, [11,23] and one study in a Chinese population reported that hearing loss was positively correlated with low HDL, hyperglycemia, and central obesity [7].

Several studies also observed relationships between obesity-related anthropometric indices and hearing loss. For example, one study found that WHR may be a surrogate marker for predicting the risk of hearing loss, [18] whereas another study suggested that FRAs were associated with hearing loss at specific frequencies, as determined by sex and the presence of diabetes, and that visceral adipose tissue (VAT) is particularly important role for hearing [25]. Two studies found relationships between BMI and hearing loss, with one finding that underweight and severe obesity were associated with an increased prevalence of hearing loss in a Korean population, and the other reporting that overweight was associated with an increased risk of hearing loss in a Japanese population [26].

This study had several limitations. First, because it was a cross-sectional study, the causative relationships between hearing loss and components of metabolic syndrome could not be determined. Second, because these data were from health examinations, the correlation between the prevalence of metabolic syndrome and hearing loss could not be accurately determined. Third, this study did not evaluate subjects who received medical treatment for hearing loss, nor did it evaluate factors contributing to hearing loss, such as noise, ototoxic drugs, otitis media, and family history of hearing loss.

## 5. Conclusions

This cross-sectional study involving a large population analyzed the association between metabolic syndrome and hearing loss. Hearing loss showed a positive correlation with the number of diagnostic factors for metabolic syndrome, especially in subjects with four specific factors: high waist circumference, blood pressure, triglyceride concentration, and fasting blood glucose concentration.

This study highlights the importance of control of metabolic syndrome in management of hearing loss. Subjects with metabolic diseases should therefore undergo regular hearing tests and, if necessary, hearing rehabilitation along with the management and treatment of their metabolic diseases. In addition, patients and health professionals may not be aware of this information regarding association between metabolic syndrome and hearing loss; hence, this can be included in part of health education.

## Figures and Tables

**Figure 1 jcm-10-04866-f001:**
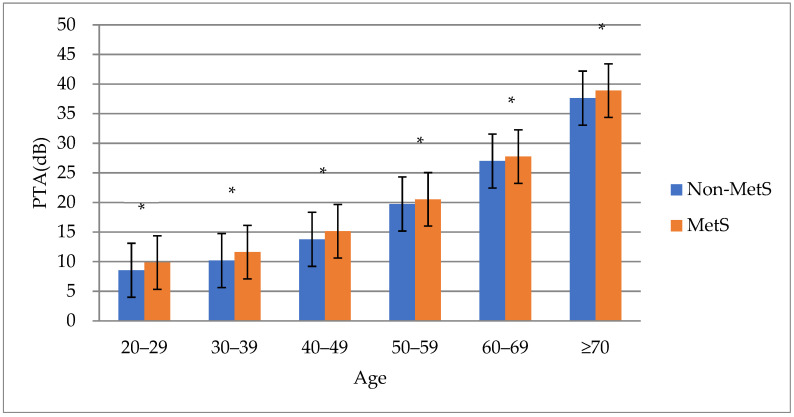
PTA threshold according to age. Abbreviations: MetS, metabolic syndrome; Non–MetS, without metabolic syndrome; PTA, pure tone audiometry. * Statistically significant. *p* < 0.05.

**Table 1 jcm-10-04866-t001:** Studies assessing the association between hearing loss and metabolic syndrome.

Author (Year)	Study Design	N	Age (Years)	Associated Variables	Conclusions
Kang S.H. et al. (2015) [9]	Cross-sectional	16,554	50.4 ± 16.6 (men) 49.2 ± 16.4 (women)	MetS or CKD/Hearing thresholds	MetS is associated with hearing thresholds in women; and CKD is associated with hearing thresholds in men and women. Subjects with MetS or CKD should be closely monitored for hearing impairment.
Sun Y.-S. et al. (2015) [3]	Cross-sectional	2100	≤65	MetS components/SNHL	Significant associations between the number of components of metabolic syndrome and hearing thresholds in US adults, with the strongest association between low HDL and hearing loss.
Kang S.H. et al. (2015) [18]	Cross-sectional	8198	54.7 ± 9.9	WHR/HL	WHR may be a surrogate marker for predicting the risk of hearing loss resulting from metabolic syndrome.
Bener A. et al. (2016) [19]	Cross-sectional	459	20–59	DM, HTN/HL	Adults with DM and hypertension showed greater hearing impairment in a highly endogamous population. Diabetic patients with hearing loss were likely to have high blood glucose and other risk factors like hypertension, retinopathy, nephropathy, and neuropathy.
Kim S.H. et al. (2016) [20]	Cross-sectional	61,052	42.33 ± 7.49 (normal) 49.89 ± 9.32 (HL)	BMI/HL	Underweight and severe obesity were associated with an increased prevalence of hearing loss in a Korean population.
Kang S.H. et al. (2016) [21]	Cross-sectional	7449	53.2 ± 10.7 56.7 ± 11.0 59.8 ± 10.8	HbA1c/HL	HbA1c level was associated with hearing impairment in nondiabetic individuals.
Lee H.Y. et al. (2016) [22]	Retrospective	16,779	≥19	MetS components/SNHL	Metabolic syndrome itself was not an independent risk factor for hearing impairment. Among its individual components, only increased fasting plasma glucose was independently associated with hearing impairment.
Aghazadeh-Attari J. et al.(2017) [12]	Cross-sectional	11,114	20–60	MetS components/SNHL	Possible associations between different components of MetS (obesity, hypertension, hypertriglyceridemia, high fasting glucose levels, and waist circumference) and SNHL in a population of West Azerbaijan drivers.
Jung D.J. et al. (2017) [23]	Cross-sectional	18,004	>40	TG/HDL ratio/SNHL	High TG/HDL-C ratio was associated with hearing impairment in a Korean population
Nwosu J. N. et al. (2017) [24]	Case-control	416	26–80 years	DM/HL	High prevalence of hearing loss among diabetic adults at University of Nigeria Teaching Hospital, Enugu. Hearing loss was predominantly sensorineural and often mild to moderate in severity.
Lee Y. et al. (2017) [25]	Retrospective	2602	57.6 ± 7.3	Factors relevant abdominal fats (FRAs)/ Age-related hearing loss (ARHL)	FRAs were associated with frequency-specific hearing losses according to sex. DM and visceral adipose tissue (VAT) are particularly important role for hearing.
Kim T.S. et al. (2017) [8]	Prospective	1381	>50	MetS components/ age-related hearing impairment (ARHI)	MetS is associated with age-related hearing impairment in women aged ≥ 50 years. At 5-year follow-up, high-frequency hearing loss tended to be greater in women with than without MetS, suggesting the need for hearing evaluation in older women with MetS.
Han X. et al. (2018) [7]	Cross-sectional	18,824	61.1 ± 7.6 66.6 ± 7.2 71.0 ± 7.7	MetS components/SNHL	MetS, including its components central obesity, hyperglycemia, and low HDL-C levels, is positively associated with hearing loss.
Jung D.J. et al. (2019) [11]	Cross-sectional	17,513,555	>40	MetS components/SNHL	Among the components of MetS, low HDL and high TG levels were especially associated with hearing loss. Rather than assessing MetS, each MetS component should be evaluated individually.
Shim H.S. et al. (2019) [10]	Cross-sectional	28,866	all age groups	MetS components/SNHL	MetS may be associated with hearing loss, especially in subjects who meet four or five of the diagnostic criteria for MetS.
Hu H. et al. (2020) [26]	Prospective cohort	48,549	20–64	BMI (w/o Waist circumference)/SNHL	Overweight and obesity are associated with an increased risk of hearing loss, with metabolically unhealthy status conferring an additional risk.
Kim J. et al. (2021) [27]	Cross-sectional	10,356	40–80	MetS components/HL	MetS is associated with high-frequency hearing loss in subjects exposed to noise.

Abbreviations: MetS, metabolic syndrome; CKD, chronic kidney disease; SNHL, sensorineural hearing loss; HL, hearing loss, DM, diabetes mellitus; HTN, hypertension; BMI, body mass index; WHR, waist hip ratio; HDL, high density lipoprotein; TG, triglyceride.

**Table 2 jcm-10-04866-t002:** Demographic characteristics of study subjects.

		Non-MetS		MetS		*p*-Value
		*n* or Mean	% or SD	*n* or Mean	% or SD	
Age		43.91	9.42	46.12	9.30	<0.0001 *^,a^
Sex	Male	45,157	(54.53%)	8253	(72.31%)	<0.0001 *^,b^
	Female	37,652	(45.47%)	3161	(27.69%)	
Waist circumference		82.00	7.86	93.40	8.15	<0.0001 *^,a^
Systolic blood pressure		116.39	11.18	127.91	11.97	<0.0001 *^,a^
Diastolic blood pressure		70.96	8.99	79.11	9.29	<0.0001 *^,a^
HDL		61.84	15.62	45.79	11.83	<0.0001 *^,a^
TG		107.12	64.92	233.77	141.82	<0.0001 *^,a^
FPG		90.89	14.14	109.39	28.33	<0.0001 *^,a^
BMI		23.26	3.00	27.50	3.24	<0.0001 *^,a^

Abbreviations: MetS, metabolic syndrome; Non-MetS, without metabolic syndrome; HDL, high density lipoprotein cholesterol; TG, triglyceride; FPG, fasting plasma glucose; BMI, body mass index. * *p* values between Non-MetS and MetS were tested using the t-test ^a^ and chi-square ^b^. * Statistically significant. *p* < 0.05.

**Table 3 jcm-10-04866-t003:** Comparison of hearing by age groups in MetS and non-MetS subjects.

Age (Years)	Non-MetS					MetS					*p*-Value ^a^
	Normal		Hearing Loss		Total	Normal		Hearing Loss		Total	
20–29	4645	(99.06%)	44	(0.94%)	4689	272	(98.19%)	5	(1.81%)	277	0.1943
30–39	20,996	(97.78%)	477	(2.22%)	21,473	2309	(96.81%)	76	(3.19%)	2385	0.0041 *
40–49	32,068	(92.98%)	2422	(7.02%)	34,490	4465	(91.48%)	416	(8.52%)	4881	0.0002 *
50–59	14,199	(79.26%)	3715	(20.74%)	17,914	2316	(76.84%)	698	(23.16%)	3014	0.0028 *
60–69	1828	(56.86%)	1387	(43.14%)	3215	341	(53.36%)	298	(46.64%)	639	0.1064
≥70	197	(29.1%)	480	(70.9%)	677	40	(23.53%)	130	(76.47%)	170	0.1532

Abbreviations: MetS: metabolic syndrome; Non-MetS, without metabolic syndrome. * *p* values between Non-MetS and MetS were tested using the Fisher’s exact test ^a^. * Statistically significant. *p* < 0.05.

**Table 4 jcm-10-04866-t004:** Comparison of the groups of subjects with normal hearing and hearing loss.

	Subject Group						
	Normal Hearing (*n*)			Hearing Loss (*n*)			*p*-Value ^a,b^
	*n* (%)	PTA(dB)	SD	*n* (%)	PTA(dB)	SD	
Non-MetS	38,069 (92.0%)	11.11	4.96	3293 (7.9%)	38.70	15.44	<0.0001 *
Non-MetS (1 factor)	22,635 (87.9%)	12.25	5.22	3109 (12.1%)	38.72	14.75	<0.0001 *
Non-MetS (2 factors)	13,272 (86.2%)	12.84	5.23	2123(13.8%)	38.16	13.83	<0.0001 *
MetS (3 factors)	6900 (86.2%)	13.19	5.20	1098 (13.8%)	37.79	13.56	<0.0001 *
MetS (4 factors)	2396 (84.5%)	13.34	5.12	438 (15.5%)	37.56	13.56	<0.0001 *
MetS (5 factors)	449 (83.7%)	13.65	5.28	87 (16.3%)	38.86	12.95	<0.0001 *

Results are reported as mean ± SD or as *n* (%). Abbreviations: MetS, metabolic syndrome; Non-MetS, without metabolic syndrome; PTA, pure tone audiometry. * *p* values between Normal hearing and Hearing loss were tested using the t-test ^a^ and chi-square ^b^. * Statistically significant. *p* < 0.05.

**Table 5 jcm-10-04866-t005:** Crude and adjusted odds ratios of hearing loss among subjects with metabolic syndrome.

	Univariable Analysis				Multivariable Analysis ^a^			
	OR	95% CI		*p*-Value ^a^	OR	95% CI		*p*-Value ^b^
Sex	1.97	1.89	2.06	<0.0001 *	1.89	1.80	1.99	<0.0001 *
Age	1.13	1.13	1.14	<0.0001 *	1.13	1.13	1.14	<0.0001 *
MetS (3 factors)	1.38	1.29	1.48	<0.0001 *	1.06	0.99	1.14	0.1119
MetS (4 factors)	1.59	1.43	1.76	<0.0001 *	1.29	1.15	1.44	<0.0001 *
MetS (5 factors)	1.68	1.34	2.12	<0.0001 *	1.21	0.94	1.56	0.1410

Abbreviations: MetS, metabolic syndrome; OR, odds ratio; CI, confidence interval. * *p* values were tested using univariate ^a^ and multivariate ^b^ logistic regression analyses. * Statistically significant. *p* < 0.05. ^a^ Including the variables age, sex, and metabolic syndrome.

**Table 6 jcm-10-04866-t006:** Association between hearing loss and metabolic syndrome.

Risk Factors	Univariable Analysis				Multivariable Analysis			
	OR	95% CI		*p*-Value ^a^	OR	95% CI		*p*-Value ^b^
Waist circumference	0.96	0.92	1.01	0.1003	1.09	1.04	1.15	0.0009 *
Systolic BP	1.72	1.64	1.80	<0.0001 *	1.12	1.06	1.18	<0.0001 *
Triglyceride level	1.25	1.19	1.31	<0.0001 *	1.08	1.03	1.13	0.0037 *
HDL-C level	1.20	1.13	1.28	<0.0001 *	1.05	0.98	1.12	0.1641
FPG level	2.14	2.05	2.24	<0.0001 *	1.14	1.09	1.20	<0.0001 *

Abbreviations: Systolic BP, systolic blood pressure; HDL-C, high density lipoprotein cholesterol; FPG, fasting plasma glucose; OR, odds ratio; CI, confidence interval. * *p* values were tested using univariate ^a^ and multivariate ^b^ logistic regression analyses. * Statistically significant. *p* < 0.05.

## Data Availability

The data presented in this study are available on request from the corresponding author. The data are not publicly available because the health examination data of a private hospital was used.

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
