# Peer review of "Association of Metabolic Syndrome with Sensorineural Hearing Loss"

_jcm, 2021, doi:10.3390/jcm10214866_

Round 1
Reviewer 1 Report
The study by Rim et al. analyzed the database from healthy examination and investigated the association between sensorineural hearing loss and metabolic syndrome. They observed that the number of components of the metabolic syndrome is positively correlated with the rate of sensorineural hearing loss, which was defined as PTA > 26 dB HL. In general, the manuscript is well written. I have few suggestions about data shown in the manuscript which the authors could probably provide or explain to improve the paper and support the conclusion.
- In the first paragraph of results section and Table 1, the authors showed that "the average pure tone hearing thresholds were 16.7 dB in subjects with and 14.5 dB in subjects without metabolic syndrome". The authors also stated "The mean pure tone hearing threshold was found to be higher in subjects with metabolic syndrome.....That is, hearing loss was positively correlated with the occurrence of metabolic syndrome." in the discussion section. However, even in metabolic syndrome group, the PTA about 16.7 dB is not thought as a hearing loss in the clinic. As the author defined, usually hearing loss is defined as PTA > 26 dB HL.
- In the second paragraph of results section, the authors described that "the average pure tone hearing thresholds were significantly higher in subjects with than without metabolic syndrome in all age groups". Since they divide the subjects into six groups by age, 20s, 30s, 40s, 50s, 60s, and 70s, could the authors design a table to demonstrate the data of “the average pure tone hearing thresholds in subjects with than without metabolic syndrome"in each age?
- Also in the same paragraph, "comparisons by age group confirmed that the percentages of subjects with mild and moderate hearing loss were significantly higher in subjects with than without metabolic syndrome" was mentioned. Could the author show these data in Table?
- In Table 2, the PTA seems around 38 dB in the hearing loss group and no differences were observed between the patients with no or higher numbers of MetS factors. The result is different from the data from Korean database shown in Kang SH et al (2015), showing that PTA increased by the component of Metabolic syndrome. How do the authors explain it?
- Introduction section: “Studies have recently reported that metabolic syndrome may be associated with hearing loss.” Please add the references.
Reviewer 2 Report
Many thanks for asking me to review the paper. I congratulate the authors for conducting such a big study. I have some concerns and I have listed below.
Best wishes
Abtsract: Please explain metabolic syndrome in abstract section
Line 50: Methods – No mention regarding ethical approval or approval form local research ethic committee.
Line 57: should this read as “This study included a total of 94,223 people, ranging in age from 20s to 70s” instead of “This study included a total of 94,223 people, ranging in age from their ranged 20s to their 70s”
Line 80: calculating their sum using the equation: (500 80 Hz + 2x1000 Hz + 2x2000 Hz + 4000 Hz)/6 – Please provide reference. Please explain, why this formula was chosen instead of averaging the audiometry results?
Line 107: only 12.1% had metabolic syndrome in the study? Could this have skewed the results?
Line: 124 and line 133
Rates of hearing loss in subjects with 0, 1, 2, 3, 4, and 5 of the components of metabolic syndrome were 7.9%, 12.1%, 13.8%, 13.8%, 15.5% and 16.3%.
Relative to subjects with 0 components of metabolic syndrome, the odds ratios of hearing loss in subjects with 3, 4 and 5 components of metabolic syndrome were 1.06 (p=0.1119), 1.29. (p<0.0001), and 1.21 (p=0.1410), respectively ><0.0001) and 1.21(p=0.1410)
It can be noted that the rates of hearing loss is not much between 2,3 components and 4, 5 components of metabolic syndrome. The hearing loss is not statistically significant with 3 and 5 components. It is seen in table 2, the hearing loss is about 37 and 39 db between non-met and met of any components. As a reader I feel that the hearing loss in not much between non-met and met group. How did the authors conclude that met group has more hearing loss?
Line 162 : Table 5 – very impressive , immense effort has gone to collect the information
Results table: It will be useful to know where you have used t-test and chi square instead of just showing the p-value.
Conclusions
- You can discuss the wider implications of this study.
- Patients with metabolic syndrome could be offered screening for hearing as a part of their general health checks.
- The study also highlights the importance of control of metabolic syndrome in management of hearing loss.
- Patients and health professionals may not be aware of this information regarding association between metabolic syndrome and hearing loss, hence this can be included in part of health education .

Round 2
Reviewer 1 Report
The authors have answered my questions and improved the manuscript. However, I have some suggestions and comment about the data presented in the result section:
- I suggest the authors provide the table or figure (with legends) to reveal the data about the average pure tone hearing thresholds in subjects with than without metabolic syndrome in each age. I don’t think that the Journal of Clinical Medicine constrains the space to place the table or figure. If the journal does constrain the space, the table or figure can at least be put in the supplementary materials.
- The table provided in response 3 seems not to answer my point 3 well to show the data of “comparisons by age group confirmed that the percentages of subjects with mild and moderate hearing loss were significantly higher in subjects with than without metabolic syndrome (p<0.001).”
Author Response
We would like to thank the editor and reviewers for their helpful comments and suggestions. We have revised our manuscript in response to these comments, and our point-by-point responses to the suggestions of the reviewers are provided below.
Response to Reviewer 1 Comments
Point 1. I suggest the authors provide the table or figure (with legends) to reveal the data about the average pure tone hearing thresholds in subjects with than without metabolic syndrome in each age. I don’t think that the Journal of Clinical Medicine constrains the space to place the table or figure. If the journal does constrain the space, the table or figure can at least be put in the supplementary materials.
Response 1 : As recommended, We have added the following Figures to manuscript. (Please see the attached file)
Point 2. The table provided in response 3 seems not to answer my point 3 well to show the data of “comparisons by age group confirmed that the percentages of subjects with mild and moderate hearing loss were significantly higher in subjects with than without metabolic syndrome (p<0.001).”
Response 2 : Unfortunately, it was revealed through the helpful comment of the reviewer that there was a problem in the table prepared by the statistical analysis of this study. To compensate for the problem, a new table was created through appropriate statistical analysis. (Please see the attached file)
According to the results of the table, “comparisons by age group confirmed that the percentages of subjects with mild and moderate hearing loss were significantly higher in subjects with than without metabolic syndrome (p<0.001).” need to be modified as “comparisons by age group confirmed that the percentages of subjects with hearing loss was higher in subjects with metabolic syndrome than in subjects without metabolic syndrome, and there was a significant statistical difference in the groups in their 30s, 40s, and 50s (p<0.05).”
These corrections and Table 2 are included in the revision of the manuscript.

Reviewer 2 Report
I like to thank the authors for taking time and effort to consider the suggestions. Once again very good work, I have put my comments below.
Best wishes
Rebuttal
We would like to thank the editor and reviewers for their helpful comments and suggestions. We have revised our manuscript in response to these comments, and our point-by-point responses to the suggestions of the reviewers are provided below.
Response to Reviewer 2 Comments
Point 1. Abtsract: Please explain metabolic syndrome in abstract section
Response 1 : As recommended. We have added the following explanation of metabolic syndrome to the Abstract : “Metabolic syndrome(MetS) is a cluster of co-occurring conditions that increase the risk of heart disease, stroke and type 2 diabetes, along with other conditions that affect the blood vessels. Components of MetS include insulin resistance, body weight, lipid concentration, blood pressure, and blood glucose concentration, as well as other features of insulin resistance such as microalbuminuria.”
Reviewer response: Agree, thank you
Point 2. Line 50: Methods – No mention regarding ethical approval or approval form local research ethic committee.
Response 2 : As recommended, we have added a statement regarding ethical approval to the Method section.
“The study protocol was approved by the Institutional Review Board of Kyung Hee University Medical Center (KMC 2019-07-065).”
Reviewer response: Agree, thank you
Point 3. Line 57: should this read as “This study included a total of 94,223 people, ranging in age from 20s to 70s” instead of “This study included a total of 94,223 people, ranging in age ranging in age from 20s to 70s”
Response 3 : As recommended, we have changed the phrase “ranging in age from their ranged 20s to their 70s” to “ranging in age from 20s to 70s”.
Reviewer response: Agree, thank you
Point 4. Line 80: calculating their sum using the equation: (500 80 Hz + 2x1000 Hz + 2x2000 Hz + 4000 Hz)/6 – Please provide reference. Please explain, why this formula was chosen instead of averaging the audiometry results?
Response 4 :
We used the six-division method for NIHL diagnosis given in the criteria of the Enforcement Decree of the Industrial Accident Compensation Insurance Act. We have added the following four relevant references:
- Kim K.S. A Review of Occupational Disease Certification Criteria for Noise-Induced Hearing Impairment. Audiol Speech Res 2017;13(4):265-271.
- Lee Y.; Park S.; Lee S.J. Exploring Factors Related to Self-perceived Hearing Handicap in the Elderly with Moderate to Moderately-severe Hearing Loss. Commun Sci Disord 2020; 25(1): 142-155.
- Shin J.W.; Kim S.W.; Kim Y.W.; Jang W.; Kim B.H.; Lim Y.S.; Park S.W.; Cho C.G. The Value of Posterior Semicircular Canal Function in Predicting Hearing Recovery of Sudden Sensorineural Hearing Loss. Res Vestib Sci Vol. 18, No. 4, Dec. 2019, 103-110
- Kim S.L.; Oh S.J.; Kong S.K.; Goh E.K. Sudden Sensorineural Hearing Loss after Granulocyte-Colony Stimulating Factor Administration. J Clinical Otolaryngol 2018;29:87-90.
Reviewer response: Agree, thank you
Point 5. Line 107: only 12.1% had metabolic syndrome in the study? Could this have skewed the results?
Response 5 :
This study is based on health examination data for relatively healthy patients. In this study, 12.1% or 11,413 patients met the diagnostic criteria for metabolic disease. First, due to the nature of this study, we cannot rule out the possibility that the study population was biased toward people interested in health. In addition, it is possible that there were fewer subjects who met the criteria for metabolic syndrome because at the time of assessment subjects may have been taking medications to control blood pressure, blood sugar, and/or other blood levels. However, I think that the results of this study are meaningful because they included a sufficient number of subjects.
Reviewer response: Agree, thank you, can you please mention this in the discussion.
Point 6. Line: 124 and line 133
Rates of hearing loss in subjects with 0, 1, 2, 3, 4, and 5 of the components of metabolic syndrome were 7.9%, 12.1%, 13.8%, 13.8%, 15.5% and 16.3%.
Relative to subjects with 0 components of metabolic syndrome, the odds ratios of hearing loss in subjects with 3, 4 and 5 components of metabolic syndrome were 1.06 (p=0.1119), 1.29. (p<0.0001), and 1.21 (p=0.1410), respectively ><0.0001) and 1.21(p=0.1410)
It can be noted that the rates of hearing loss is not much between 2,3 components and 4, 5 components of metabolic syndrome. The hearing loss is not statistically significant with 3 and 5 components. It is seen in table 2, the hearing loss is about 37 and 39 db between non-met and met of any components. As a reader I feel that the hearing loss in not much between non-met and met group. How did the authors conclude that met group has more hearing loss?
Response 6 : This study had several limitations,; however we believe that the study demonstrated an association between hearing loss and metabolic syndrome.
The threshold of PTA was significantly higher in the group with metabolic syndrome in Table 1 and the rate of hearing loss was also higher in the group with metabolic syndrome in Table 2. In addition, Table 3 shows that although statistical significance was not observed for 3 and 5 factors in the multivariable analysis, this can be attributed to a decrease in the number of relevant groups. and the odds ratio of hearing loss tends to increas according to the number of metabolic syndrome factors.
Hence, when evaluating the effect of metabolic syndrome in hearing loss, we should evaluate each component individually rather than the diagnosis of metabolic syndrome itself and the number of components of metabolic syndrome.
Reviewer response: Please discuss this in the discussion section. The other possibility is that hearing loss can be because of multiple un explained factors and further research may possibly give us more answers
Point 7. Results table: It will be useful to know where you have used t-test and chi square instead of just showing the p-value.
Response 7 : We have described which test (t-test or chi square) was used in the text.
Reviewer response: Agree, thank you. It will be helpful if authors also mention in the table for the ease of reader.
Point 8. Conclusions
You can discuss the wider implications of this study. Patients with metabolic syndrome could be offered screening for hearing as a part of their general health checks. The study also highlights the importance of control of metabolic syndrome in management of hearing loss. Patients and health professionals may not be aware of this information regarding association between metabolic syndrome and hearing loss, hence this can be included in part of health education.
Response 8 : Thank you for your comment. The feedback of the reviewer is reflected in the conclusion of the revised manuscript.
Reviewer response: Agree, thank you

Author Response
We would like to thank the editor and reviewers for their helpful comments and suggestions. We have revised our manuscript in response to these comments, and our point-by-point responses to the suggestions of the reviewers are provided below.
Response to Reviewer 2 Comments
Point 5. Reviewer response: Agree, thank you, can you please mention this in the discussion.
Response :
We mentioned the content of point 5 in the discussion.
Point 6. Reviewer response: Please discuss this in the discussion section. The other possibility is that hearing loss can be because of multiple unexplained factors and further research may possibly give us more answers
Response :
We discussed the content of point 6 in the discussion.
Point 7. Reviewer response: Agree, thank you. It will be helpful if authors also mention in the table for the ease of reader.
Response : We have also mentioned which test (t-test or chi square) was used in the table.
